# DIFFERENTIALLY PRIVATE CONDITIONAL TEXT GENERATION FOR SYNTHETIC DATA

## ABSTRACT

Companies have faced increasing pressure in recent years to anonymize user collected data when sharing internally or to third parties. Text data in particular contains copious amounts of personally identifiable information that has proven to be difficult to de-identify while remain useful for the party of interest. Previous works have suggested that synthetic text generation could provide a promising avenue to curate high performant and private datasets. In this paper, we introduce an approach to synthesize high utility text classification datasets by performing conditional generation through a large language model, distilGPT2, while providing measurable guarantees via differential privacy. We show that naive approaches suffer heavily from utility loss by entangling task-relevant factors in the transformer embedding space, making controlled generation more difficult. We analyze how incorporating a secondary learning objective can improve the performance of the generative model, improving utility of the generated data.

## 1 INTRODUCTION

In recent years, language models have seen dramatic improvements in performance over NLP tasks. In large part, this has been due to the rapid accumulation of user generated text on the internet. Companies have been able to aggregate millions of documents available online as well as their user data to train these large language models. However, lawmakers and their constituents have grown wary of data collection and usage practices, urging more stringent regulations.

In 2018, the EU set the General Data Protection Regulation (GDPR) into motion, with the goal to increase transparency about collected information and give users more control over how their data is handled. (Voigt & Bussche, 2017). Consequently, companies are now searching for ways to utilize user data without exploiting user privacy. The GDPR begins with the statement: "The protection of natural persons in relation to the processing of personal data is a fundamental right"; it is imperative that we innovate on methods to use data effectively without risking user privacy.

In this paper, we study privatization of unstructured text data. Even with safety measures in mind, there has been massive exploitation of user text data. For example, in 2006, as part of their algorithm contest, Netflix released a de-identified dataset of user generated movie reviews. Researchers discovered that surprisingly little information was required to reconstruct the identities of users that contributed to the reviews (Narayanan & Shmatikov, 2006). Further studies have shown how other methods, such as authorship and membership inference attacks (Carlini et al., 2020), can be utilized to reconstruct user identities. All this to say, without proper privacy guarantees and careful data analysis, companies risk user data to exploitation.

Dwork (2006) and Abadi et al. (2016) proposed differential privacy (DP) and DP-SGD/DP-Adam, respectively, as methods to provide provable and quantifiable guarantees about privacy. Generally, we say that a randomized algorithm satisfies DP if the output distribution is indistinguisable when run on neighboring datasets. However, current trade-offs between privacy and utility, particularly in synthetic text generation, makes it impractical for companies to create useful data with strong privacy guarantees.

A common approach for anonymization is to de-identify (redact) personally identifiable tokens in text, such as names and addresses. While this may seem like a reasonable approach on paper with SOTA models reporting accuracies of nearly than 97%, the 3% of tokens that are misidentified could

be used by an adversary to re-identify users. Consequently, this approach isn't a strong enough guarantee of privacy. A permissible error from such a model should be lower than 1% (Yogarajan et al., 2020; Al Aziz et al., 2021), something that has not been achieved today for abitrary datasets. Synthetic data is promising because it avoids the problem of anonymizing an individual's data by instead producing information about non-existent persons.

Other approaches to anonymize unstructured text data have focused on word or sentence level perturbations in order to reduce vulnerability to membership inference and authorship attacks. These approaches often heavily degrade semantic quality of the text and may struggle to provide overall privacy guarantees in the context of language peculiarities, such as with the leakage of PII. Other approaches seek to generate data synthetically, such as Libbi et al. (2021) and Al Aziz et al. (2021). However, such studies often show a large tradeoff between privacy and utility or make differentially private guarantees with a potentially unreasonable epsilon parameter (e.g. $\epsilon > 10$).

In this paper, we present an approach of generating synthetic text data by performing controllable generation through a large language model. We show it is possible to synthesize text classification datasets with rigorous privacy guarantees. We hope this method will enable companies to share data and train high utility models without putting their users' data at risk. Our contributions are as follows:

1. We present findings on problems that arise when performing conditional finetuning of large language models with DP-Adam. Particulary, we find that it becomes difficult to conditionally prompt the model towards a desired class and generate synthetic data that mimics desired attributes of the original. We propose using a task-relevant loss via a secondary learning objective to solve this issue.

2. We generate synthetic versions of the SST-2 and AG News datasets by performing conditional text generation over a langauge model. We incorporate a combination of generation techniques: attribute conditioning and a gradient based approach (Dathathri et al., 2019) to further steer generation. We show minimal loss in utility of our synthetic datasets (6.3%) with strong privacy guarantees ($\epsilon = 3$).

Code to recreate our results are available here: (redacted for review)

## 2 BACKGROUND

### 2.1 LANGUAGE MODELING

Given a sequence of tokens X = $x_0$, ... , $x_n$ , language models (LMs) are trained to compute the unconditional probability of the sequence p(X). This probability can be rewritten in terms of product of conditional probabilities by recursively applying the chain-rule (Bengio et al., 2003) as:

$$p(X) = \prod_{i=1}^{N} p(x_i|x_0, ..., x_{i-1}) \tag{1}$$

This allows modeling the language via next-word prediction. We use the transformer architecture (Vaswani et al., 2017) to model the distribution of natural language. Generation of a new sequence $y$ can be created by sequentially sampling its constituents: $p_\theta(y_0), p_\theta(y_1|y_0), ..., p_\theta(y_m|y_{<m})$.

### 2.2 CONDITIONAL TEXT GENERATION

Conditional generation of text attempts to steer the output of a LM given a desired condition or control variable. Keskar et al. (2019) introduced a method to accomplish this goal by performing training a LM over a dataset, such that the desired condition is prepended to the text body: "BOS [condition] SEP text" (BOS and SEP are special tokens to indiciate the beginning of the sentence and to separate label from the text body, respectively).

On the other hand, plug and play controllable language generation (PPLM) (Dathathri et al., 2019) combines an attribute model (such as a discriminator) with a LM to manipulate its output and perform controllable text generation. Given an attribute $a$ and generated text $x$, let the output of the

discriminator model represent $p(a|x)$. In order to control generation, we shift the latent hidden state of the language model at step $i$, $h_i$ by $\Delta h_i$ in the direction of the sum of two gradients: (1) towards a smaller cross entropy loss in the attribute model $p(a|x)$ for the desired attribute $a$ and (2) toward higher log likelihood of the language modeling head $p(x)$ to preserve the generation quality and fluency.

In this paper, we use a combination of the two approaches in order to generate high-quality data. We first fine-tune a large language model over the desired dataset with conditional prompting similar to Keskar et al. (2019) and then use the gradient-based approach as described by Dathathri et al. (2019) to steer generation with high likelihood towards the desired attribute. With this process, we can generate labeled data for our synthetic dataset.

## 2.3 DIFFERENTIAL PRIVACY

Differential Privacy (DP) is a formal definition of privacy which offers strong assurances against various re-identification and re-construction attacks (Dwork, 2006; Dwork & Roth, 2013). In recent years, DP has attracted significant attention due to its mathematically sound and provable privacy guarantees. Moreover, it has unique properties such as robustness to auxillary information and post-processing, composability to enable modular design, and group privacy. (Dwork & Roth, 2013; Abadi et al., 2016).

**Definition 1.** *(Differential Privacy (Dwork, 2006)) A randomized function $\mathcal{M}$ provides $(\epsilon, \delta)$-differential privacy if for all adjacent datasets $X, X' \in \mathcal{X}$ and all $Y \subset \mathcal{Y}$,*

$$Pr[\mathcal{M}(X) \in Y] \leq \exp(\epsilon) \cdot Pr[\mathcal{M}(X') \in Y] + \delta \tag{2}$$

This is a standard definition of DP, which implies that the outputs of a DP model/algorithm for neighboring datasets are indistinguishable, bounded by the privacy parameter $\epsilon$. $\epsilon$ is a non-negative number which represents the privacy budget. Smaller $\epsilon$ values more rigorously enforce privacy, but may have the effect of decreasing data utility. DP also allows for tracking privacy loss throughout the execution of a program by computing its leakage parameters. In this paper, we use Renyi Differential Privacy for accounting privacy budget (Mironov, 2017).

Composability and robustness to post-processing are important properties of DP that are necessary for the guarantees in our paper. Composability allows for reasoning about overall privacy loss from the composition of multiple DP algorithms releasing multiple statistics about a particular dataset. Robustness to post-processing implies that if some mechanism $\mathcal{M}$ satisfies $\epsilon$-differential privacy, then for any deterministic or randomized function $\mathcal{F}$, so does $\mathcal{F}(\mathcal{M})$. This allows us to make $\epsilon$-DP guarantees about the generated text from our $\epsilon$-DP trained language model.

**Definition 2.** *Differentially Private Stochastic Gradient Descent (DP-SGD) modifies the update step during backpropagation by (1) clipping the gradient for each example in the mini-batch to a maximal norm $C$ and (2) adding Gaussian noise with standard deviation proportional to $C$ to the mean of the clipped gradients.*

$$w^{(t+1)} = w^{(t)} - \eta_t \cdot \frac{1}{B}\{\sum_{i \in \mathcal{B}_t} clip_C(\nabla \mathcal{L}_i(w_t)) + N(0, \sigma^2 C^2 I)\} \tag{3}$$

Where $clip_C = v \cdot \min(1, \frac{C}{||v||_2})$. Intuitively, the DP-SGD mechanism preserves privacy by mitigating the impact of out-of-distribution samples on the model, and is used during fine-tuning of our language models. DP-Adam is the differentially private version of the Adam optimizer (Kingma & Ba, 2014), using the same gradient privitization as outlined in DP-SGD.

## 3 RELATED WORKS

Current methods on text privitization fall into three general categories: word/sentence level perturbations, private text embeddings, and synthetically generated text. Here, we discuss each method.

**Word/Sentence Level Perturbations:** Many works have discussed anonymizing text by perturbing word or sentence level embeddings to satisfy $\epsilon$-differential privacy. This set of approaches change individual words in a document, often following a variant of metric based DP (Alvim et al., 2018)

which has shown to be a more utilitarian perspective of privacy in the context of NLP. However, as discussed by Mattern et al. (2022), these perturbations struggle to provide overall privacy guarantees in the context of language peculiarities and leakage of other personally identifiable information (PII) that allow for re-identification. They also suffer from utility losses since grammatical and syntactic structure are degraded. Other methods suggested by Weggenmann & Kerschbaum (2018) and Bo et al. (2019) investigate differentially private mechanisms via latent space perturbations and adversarial training, respectively, to reduce the impact of authorship inference attacks. However, these methods, again, do not address the issue of PII leakage and suffer from significant uility losses.

**Private Text Embeddings:** Other methods have investigated releasing private text embeddings instead of the original text content. Recent work such as Lyu et al. (2020) and Xu et al. (2021) propose randomization mechanisms that can transform text embedding vectors into one that satisfies metric space differential privacy guarantees. This method has shown promise in providing formal guarantees while also retaining high utility. However, this process does not leave human readable text, which is a desired property for companies performing internal data sharing; thus, we examine our approach independent of this body of work.

**Synthetic Text:** Other methods, particularly in the medical domain, have attempted to address the issue of privacy via synthetic text generation. Synthetic data addresses the problems of de-identification by simply not describing real people, and thus retaining plausible deniability over the data produced. Recent methods like Libbi et al. (2021) and Al Aziz et al. (2021) have proposed text generation approaches; This paper goes further, investigating the impact of a large range of parameter selection in conditional text generation and most importantly, demonstrating high utility even with strong privacy parameters (e.g. $\epsilon = 3$), something previous works have not done.

## 4 DATASETS AND PREPROCESSING

In this paper, we generate artificial datasets for text classification. We choose this task because it allows us to best compare utility and privacy in one dataset. We experiment over two datasets. Each dataset is split 80:20 for train and test. We represent datasets as $D = \{(x_i, y_i)\}_{i=1}^{n}$

### 4.1 SST-2

The SST-2 corpus consists of 11,855 movie review samples, each labeled with positive orn egative sentiment by human annotators. This dataset was perfectly balanced with each class having equal representation (Socher et al., 2013).

### 4.2 AG NEWS

The AG News corpus is a topic classification task. This dataset consists of over 120,000 samples, each labeled under a topic from: Sports, World, Business, Sci/Tech. This dataset was perfectly balanced with each topic having equal representation (Zhang et al., 2015).

## 5 EXPERIMENTS

This paper improves on existing methods for generating high-utility synthetic text data with differential privacy guarantees. Bommasani et al. (2019) argued that for successful private synthetic text data, we must have formal guarnatees of privacy and have distributional similarity to the original dataset. We achieve this by conditionally finetuning a LM (distilGPT2) over the original text data, the intuition being that we can reconstruct a similar distribution via generation. Since the model is learned privately, the post-processing theorem (Dwork, 2006) allows us to make the same $\epsilon$ guarantees about the generated samples. We show that with this approach, we are able to construct private, synthetic data that retains high utility. We hope that this will enable companies to utilize synthetic data, reducing reliance on private user information.

All our experiments were run on one NVIDIA V100 GPU instance.

### 5.1 FINE-TUNING

The baseline language model that we use for training is a pretrained distilgpt2 from HuggingFace Sanh et al. (2019). We use this model over the larger versions to provide faster iteration of training under different configurations.

We fine-tune the language model $\mathcal{G}$ to the task of synthesizing labeled sentences to obtain the fine-tuned language model $\mathcal{G}_{tuned}$. Here, $\mathcal{G}$ is specifically fine-tuned to the linguistic domain of $D_{train}$ (that is, the sentences, vocabulary, style, etc.), as well as the particular classes in $D_{train}$. The language modeling head, a feed forward network attached to the transformer architecture, is used to model the distribution of the next-word from an input sequence. During generation, we sample from this head. Generally speaking, we would like to use $\mathcal{G}_{tuned}$ to generate a sentence set of any length with conditioned attribute $a$ being the class label.

We fine-tune $\mathcal{G}$ by training it over the data from $D_{train} = \{(x_i, y_i)\}_{i=1}^n$. We generate training samples for conditional finetuning by prepending the label with the text body so that we end up with: $U = \text{BOS } y_i \text{ SEP } x_i$. We fine-tune this model under different privacy settings, specified by the epsilon parameter. When training with DP, the Adam optimizer is substituted with the DP-Adam optimizer implemented from the private-transformers library [1], provided by Li et al. (2021). We also use the ghost-clipping mechanism outlined by Li et al. (2021) which introduces a memory efficient method to perform per-example gradient clipping. Renyi differential privacy (Mironov, 2017) was used to account privacy budget during training.

#### 5.1.1 BASELINE METHOD 1: CONDITIONAL FINE-TUNING WITH FILTER

In our first approach, we (1) perform full fine-tuning of $\mathcal{G}$ with the training procedure described above to produce $\mathcal{G}_{tuned}$. (2) We independently train a discriminator to model $p(a|x)$, the probability of generated sample, $x$, to belong to the class $a$. In our work, we model the discriminator by fine-tuning a language model for classification over the dataset. (3) We conditionally generate $n_a$ samples for each class $a$ from $\mathcal{G}$ and filter out any samples that do not meet a desired threshold score from the discriminator (e.g. only include the sample if $p(a|x) > 0.5$). Specifically, generation was done by performing nucleus sampling (Holtzman et al., 2019) over the output distribution of $\mathcal{G}_{tuned}$. The described approach is similar to several methods used in data augmentation (Anaby-Tavor et al., 2019; Bayer et al., 2022; Queiroz Abonizio & Barbon Junior, 2020).

This approach worked well for generating artificial datasets for SST-2 and AG News in the non-private setting. We synthesized datasets for each by generating the same number of samples for each class as the original. Generation was done by simply prompting the model with "BOS class SEP". In the private setting, we replaced the Adam optimizer with DP-Adam and tracked the total privacy budget with the RDP accountant. As we improved the privacy guarantee with smaller epsilon parameters (e.g. $\epsilon = 8$), the quality of conditional generation quickly degraded. While the private LM generated text that appropriately mimicked the linguistic domain of the training data, conditional prompting did not produce consistent results; prompting the model with attribute $a$ would infrequently meet the threshold requirement from $p(a|x)$.

We also analyzed samples qualitatively and found the same results. For example, the non-private $\mathcal{G}_{tuned}$ generally produced samples that fit the class it was prompted: (e.g. "BOS positive SEP" might yield "a sensitive and heartwarming story of an aging man..."). However, the same approach with the private $\mathcal{G}_{tuned}$ produced samples that very inconsistently fit the prompted attribute (e.g. "BOS positive SEP" might yield "an emotional slap in the face, and..."). See Appendix B for more examples. Without having high confidence in our model being able to generate text conditionally for a desired class, the labels in the synthesized dataset may be meaningless. This would severely degrade the utility of the artificial data. This result suggests that a stronger mechanism than just prompting is required to steer the model towards high-quality class conditional samples.

#### 5.1.2 BASELINE METHOD 2: CONDITIONAL FINE-TUNING WITH PPLM GENERATION

Iterating from Baseline 1, we attempted to use a similar approach as PPLM (Dathathri et al., 2019), a gradient based steering mechanism, to guide the private $\mathcal{G}_{tuned}$ models towards more high qual-

---

[1] https://github.com/lxuechen/private-transformers

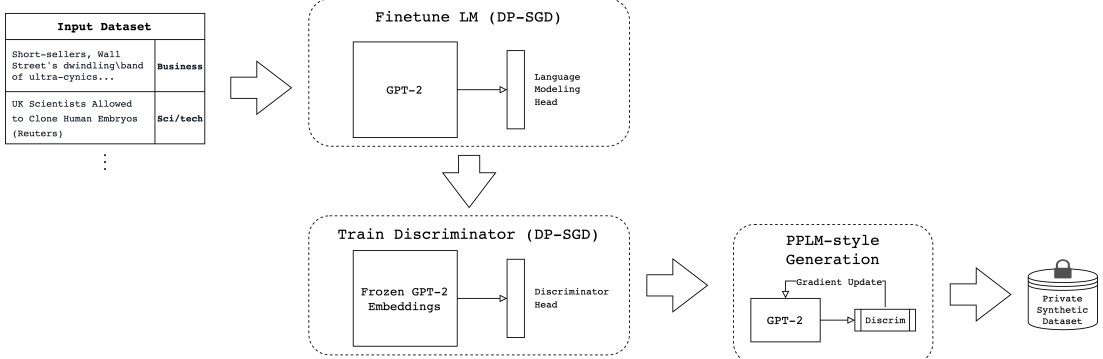

ity generation. Similar to Baseline 1, we (1) train $\mathcal{G}_{\text{tuned}}$, then (2) train a discriminator to estimate the attribute model $p(a|x)$ by training a discriminator head over the frozen $\mathcal{G}_{\text{tuned}}$ model. The discriminator head is a simple MLP with non-linear activations. Lastly, (3) we perform PPLM-based conditional generation (See Section 5.2) to generate the synthetic labeled text classification dataset.

The intuition for this approach is that the gradient based generation mechanism will guide $\mathcal{G}_{\text{tuned}}$ into generating samples that align strongly with the desired label. In order to effectively use the discriminator to perform gradient updates on the hidden states of $\mathcal{G}_{\text{tuned}}$, we trained the discriminator over the fine-tuned LM's frozen embeddings. Again, while this approach worked well in the non-private setting, it became infeasible to train the discriminator at strong epsilon settings. At $\epsilon = 3, 8$ the discriminator was not strong enough to properly contribute to generation. We hypothesized that this issue was indicative that $\mathcal{G}_{\text{tuned}}$ was not preserving information about the attribute labels during private fine-tuning, making it difficult for the discriminator to learn separation, and simulatenously making it more difficult for the LM to generate label aligned samples as observed in the previous section. We investigated this hypothesis by visualizing the embedding space of $\mathcal{G}_{\text{tuned}}$ at different epsilon settings and estimating the mutual information between the transformer embedding space and class labels by training a Random Forest classifier (See Figure 1). We hypothesize that in order to strongly reconstruct distributional properties from the original dataset, the generative model should produce embeddings that are separable with respect to those task-relevant factors.

### 5.1.3 OUR METHOD: MULTITASK CONDITIONAL FINE-TUNING WITH PPLM GENERATION

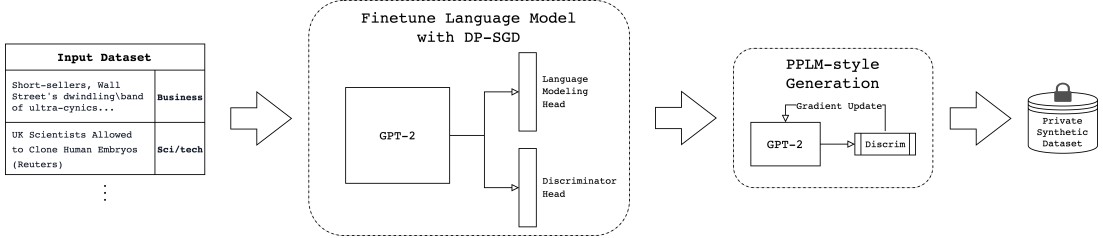

In order to address this issue we introduce a secondary learning objective and perform multitask learning during fine-tuning. In Baselines 1 and 2, the transformer is only attached to a linear language modeling head that models the probability distribution of the next word. In our approach, we simultaneously train a discriminator head, as shown in the diagram above. The discriminator head is, like Baseline 2, a simple MLP head. We now perform two gradient updates at every step – one to update the language modeling head and the other to update the discriminator head. We add the appropriate amount of noise to the gradients to maintain $\epsilon$-DP guarantees and track privacy budget throughout training with RDP (Mironov, 2017).

Since we still want to retain conditional prompting for the model, we want the language model to be able to see the conditional prompt, i.e. "BOS positive SEP text", which includes the prepended label so that the model is able to understand prompting. Meanwhile, the discriminator head should be able to learn to model $p(a|x)$ for a label, $a$, and generated sample $x$ without seeing the label in the input. So, for the language head, we feed the label prompted text data and perform a gradient

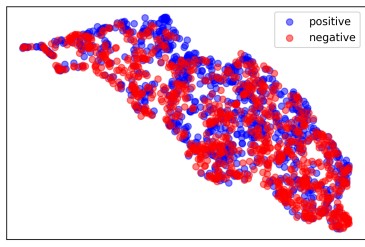

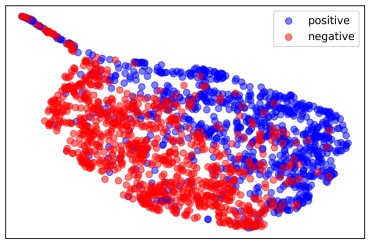

| DP Guarantee | Baseline 2 | Ours |
|---|---|---|
| $\epsilon = \inf$ | 0.803 | 0.883 |
| $\epsilon = 256$ | 0.792 | 0.873 |
| $\epsilon = 16$ | 0.773 | 0.869 |
| $\epsilon = 8$ | 0.739 | 0.865 |
| $\epsilon = 3$ | 0.693 | 0.866 |

Figure 2: Random Forest Classifier Test Accuracies over SST2 Embeddings from $\mathcal{G}_{\text{tuned}}$. The multitask approach (ours) shows marginal loss in performance at high privacy settings.

Figure 1: UMAP Projection of SST2 Embeddings from $\mathcal{G}_{\text{tuned}}$ with $\epsilon = 3$. Baseline 2 (top). Ours (bottom).

update. Then, for the discriminator head, we replace the label in the input with a random token, the intuition being that the discriminator head will pay less attention to the embeddings at that location, and be a more informative guide during generation.

We also train this discriminator head to classify text at different prefix lengths. For example, if the prefix step was specified to be 2, we would compute the loss given the transformer output for the second token, fourth token, sixth token, and so on. The loss is linearly weighted such that the first prefix is weighted the least and the last prefix is weighted the most. Lastly, this loss is averaged, and then the gradient update is computed. This loss procedure is to ensure the discriminator head is robust enough to provide meaningful classifications at different lengths of a sequence to improve its contribution during gradient based generation.

---

**Algorithm 1** DP Multitask Conditional Training

**Data:** $\mathcal{G}_{\text{pretrained}}$, $D_{\text{train}} = \{(x_i, y_i)\}_{i=1}^N$, number of iterations T, learning rates $\eta_{\text{lm}}$, $\eta_{\text{discrim}}$, noise multiplier $\sigma$, clipping bound $C$, initial parameter vectors $\theta_{\text{transf}}^{(0)}$, $\theta_{\text{lm}}^{(0)}$, $\theta_{\text{discrim}}^{(0)}$, batch size $B$, initial moment estimates $m_0, v_0 \in \mathbb{R}^p$, exponential decay rates $\beta_1, \beta_2 \in \mathbb{R}$ and constant $\gamma$

**for** $t \in [E \cdot N/B]$ **do**
    Draw batch $b_t$ from $\mathcal{D}$ with sampling probability $q$.
    **for** $(x_i, y_i) \in b_t$ **do**
        rand $\leftarrow$ random token from vocabulary
        $s_{\text{lm}} \leftarrow$ "BOS $y_i$ SEP $x_i$",    $s_{\text{discrim}} \leftarrow$ "BOS rand SEP $x_i$"
        $g_{\text{lm}}^{(t)} \leftarrow \nabla\mathcal{L}(G_{\theta_{\text{transf, lm}}^{(t)}}(s_{\text{lm}}), s_{\text{lm}})$,    $g_{\text{discrim}}^{(t)} \leftarrow \nabla\mathcal{L}(G_{\theta_{\text{transf, discrim}}^{(t)}}(s_{\text{discrim}}), y_i)$
        $g_{\text{lm}}^{(t)} \leftarrow g_{\text{lm}}^{(t)} \cdot \min(1, C/||g_{\text{lm}}^{(t)}||_2)$,    $g_{\text{discrim}}^{(t)} \leftarrow g_{\text{discrim}}^{(t)} \cdot \min(1, C/||g_{\text{discrim}}^{(t)}||_2)$
    **end**
    $g_{\text{lm}}^{(t)} \leftarrow \frac{1}{B}(\sum_{i \in b_t} g_{\text{lm}}^{(t)} + N(0, \sigma^2 C^2 I))$
    $g_{\text{discrim}}^{(t)} \leftarrow \frac{1}{B}(\sum_{i \in b_t} g_{\text{discrim}}^{(t)} + N(0, \sigma^2 C^2 I))$
    $\theta_{\text{transf, lm}}^{(t+1)} \leftarrow \text{AdamUpdate}(\theta_{\text{transf, lm}}^{(t)}, m_t, v_t, g_{\text{lm}}^{(t)}, \beta_1, \beta_2, \gamma)$
    $\theta_{\text{transf, discrim}}^{(t+1)} \leftarrow \text{AdamUpdate}(\theta_{\text{transf, discrim}}^{(t)}, m_t, v_t, g_{\text{discrim}}^{(t)}, \beta_1, \beta_2, \gamma)$

**end**

**Output:** *Trained Model* $\theta_{transf}^{(T)}, \theta_{lm}^{(T)}, \theta_{discrim}^{(T)}$

---

Ultimately, we find that by training both the discriminator and language modeling head simultaneously, $\mathcal{G}_{\text{tuned}}$ is able to conditionally generate even when trained with strong privacy guarantees. In Figure 1, we show how this approach impacts the embedding space of models trained at rigorous privacy constraints compared to the naive approach via a UMAP projection. We find that the noise injected via differential privacy doesn't prioritize the model to implicitly learn particular distributional factors about the original dataset such as separation of class labels, and an explicit loss mechanism can recover this and improve quality of generation.

## 5.2 GENERATION

Next, we describe in detail the conditional generation procedure to synthesize a private version of the dataset. We aim to generate labeled samples of text that reconstruct similar distributional properties as the original. In order to guide generation towards a particular class, we apply a PPLM (Dathathri et al., 2019) based gradient approach. We utilize the discriminator trained in the previous step to perform gradient updates over the hidden states of the model to steer the generation towards the desired class. The steps for generation of a single sample are as follows:

1. Prompt the model with **BOS class SEP** and generate the distribution of the next word via the language modeling head.
2. Compute hidden embedding states of the generated text. Pass this embedding through the discriminator, which models $p(a|x)$.
3. We now shift the hidden state, $h_i$ by summing two gradients: (1) gradient of the cross entropy loss between the discriminator output and desired class vector. (2) gradient towards the higher log likelihood of the language modeling head which models $p(x)$. This is done by minimizing the KL divergence between the modified and unmodified language mdoeling head distribution.
4. Compute the new LM head distribution from the updated latent space.
5. Sample from the new language modeling head distribution for the next word by performing nucleus sampling (Holtzman et al., 2019).
6. Repeat steps 1-3 until the termination token or the specified maximum length is reached.

We discuss further implications and limitations of this approach in Section 7.

## 6 EVALUATION

With the described approach, we generate synthetic versions of the SST-2 and AG news dataset. 5 variations are generated with different differential privacy settings: $\epsilon \in \{256, 16, 8, 3\}$ and a non-private version. The only change between the non-private and private versions are replacing the optimizer from Adam to DP-Adam provided by the private-transformers library (Li et al., 2021). The gradients in the non-private version are still clipped to the maximum gradient norm parameter, $C$.

## 6.1 PRIVACY

Differentially private training provides formal guarantees about the generated text as a consequence of the post-processing theorem. However, recent works have shown that the impact of epsilon DP on large language model training is still unclear, and we could observe empirical privacy-preservation even at high epsilon levels. To test this, we test the artificial dataset for memorization by comparing the proportion of n-grams (for $n \in [3...7]$) in the synthesized data to those present in the original dataset. Our findings are consistent with previous studies with language modeling. Empirically, we see even large epsilon settings dramatically decrease memorization in the synthesized data (Ponomareva et al., 2022).

## 6.2 UTILITY

We measure the utility of the synthetic dataset by training a classifier over the synthesized data and evaluate the performance on the held-out test dataset. We don't experiment with different classi-

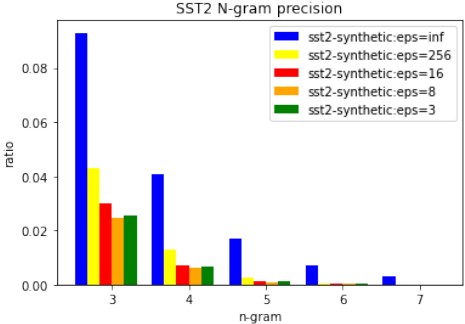

Figure 3: N-gram ratios of the different synthetic datasets trained on SST-2

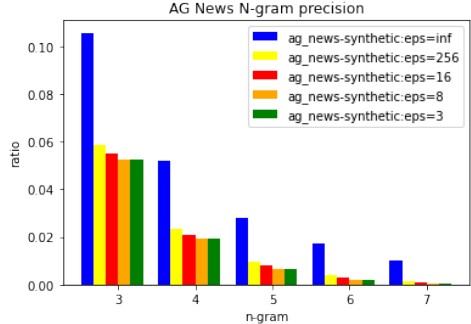

Figure 4: N-gram ratios of the different synthetic datasets trained on AG News.

fication models since our goal is to strictly evaluate the performance of the synthesized dataset. So, we choose to use a state of the art classifier, DistilBERTForSequenceClassification, from the HuggingFace transformers library.

We first train a classifier over the original dataset to produce baseline accuracies to compare the utility of the synthetic data to. Next, for each dataset variant, $\epsilon \in \{\mathrm{inf}, 256, 16, 8, 3\}$, we train a classifier. To measure the performance of the model, we compute the accuracy of the model over the held out test set. These results are shown in Table 1. We do not modify any hyperparameters of the classifier for each dataset. The selected parameters can be seen in Appendix A.

Table 1: Classification Accuracies of Our Method (Multitask Model)

| Dataset Variant | SST-2 | AG News |
|---|---|---|
| Original (baseline) | 0.941 | 0.938 |
| Synthetic Non-Private | 0.892 | 0.913 |
| Synthetic $\epsilon = 256$ | 0.883 | 0.883 |
| Synthetic $\epsilon = 16$ | 0.864 | 0.874 |
| Synthetic $\epsilon = 8$ | 0.829 | 0.871 |
| Synthetic $\epsilon = 3$ | 0.803 | 0.867 |

## 7 DISCUSSION

In this paper, we propose a method for generating synthetic text classification datasets with differential privacy guarantees by performing conditional text generation via large language models. We show the difficulties in doing this naively, particularly exploring how strong settings of privacy impact the conditional prompting scheme which has performed well in non-DP settings. By utilizing a task-relevant second learning objective and gradient based steering of generation towards a desired class, we show conditional generation is possible even at strong privacy settings. We believe this method has potential for creating synthetic datasets that will enable companies to share and train on information without putting users' personal information at risk.

However, we want to point out some limitations and future directions for this line of work. Firstly, previous studies have shown that training neural network models with DP-SGD can result in increased bias Bagdasaryan et al. (2019). In our work, we chose to use perfectly balanced datasets in order to mitigate the problems of unequal representation of classes. This could potentially lead to fairness issues when generating synthetic data, and biases from the original data may be amplified in the new dataset (Kuppam et al., 2019; Zhu et al., 2020). Future work may investigate how using this method affects fairness among groups represented in a dataset.

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

## A  HYPERPARAMETERS AND TRAINING RESULTS

Table 2: Hyperparameters for Training LM used for results in Section 7

| Method | Value |
|---|---|
| Clipping Norm $C$ | 0.1 |
| Batch Size $B$ | 128 |
| Learning rate $\eta$ | 5e-4 |
| Learning rate decay | yes |
| Epochs $E$ | 4 |
| Last $N$ Frozen Transformer Layers | 6 |
| DP Guarantee | $(\epsilon, \frac{1}{|\mathcal{D}_{\text{train}}|})$ |

Table 3: Hyperparameter search range for experiments

| Hyperparameter | Value |
|---|---|
| Clipping Norm $C$ | $\{0.1, 0.25, 0.5\}$ |
| Batch Size $B$ | $\{16, 32, 64, 128\}$ |
| Learning rate $\eta$ | $\{200, 100, 50, 10, 3\} \cdot 1e - 5$ |
| Learning rate decay | yes, no |
| Epochs $E$ | $\{3, 4, 5, 6, 7, 8\}$ |
| Last $N$ Frozen Transformer Layers | $\{2, 3, 4, 5, 6\}$ |
| DP Guarantee | $(\epsilon, \frac{1}{|\mathcal{D}_{\text{train}}|})$ |

Table 4: Test loss for SST-2 LM results in Section 7

| DP Guarantee | Loss (Naive) | Loss (Multitask) |
|---|---|---|
| $\epsilon = \inf$ | 3.823 | 3.936 |
| $\epsilon = 256$ | 3.807 | 3.729 |
| $\epsilon = 16$ | 4.033 | 3.812 |
| $\epsilon = 8$ | 4.102 | 3.883 |
| $\epsilon = 3$ | 4.183 | 3.907 |

Overall, we found that the only hyperparameters that had significant impact on the performance of the language model was learning rate and batch size, consistent with other works.

Table 5: Hyperparameters for classifier used for results in Section 7

| Hyperparameter | Value |
|---|---|
| Batch Size $B$ | 32 |
| Epochs $E$ | 5 |
| Learning rate $\eta$ | $1e-4$ |
| Last $N$ Frozen Transformer Layers | 6 |
| Learning rate decay | yes |

Table 6: Text Samples From SST2 Naive Model ($\epsilon = \inf$)

| Prompt | Text |
|---|---|
| BOS positive SEP | • a tour de force showcase that combines deep emotional connection with everyday events
• a sensitive and heartwarming story of an aging man who switches bodies to new places and lives to embrace his environment.
• cinematic poetry and soulful poetry combine together in stunning, horrifying and moving images and emotions of childhood. |
| BOS negative SEP | • frida lacks an emotional center and a strong emotional center.
• has a grisly undercurrent of a film – no humor, no suspense, no intrigue, no suspense, no suspense.
• predictably soulless and hokey, and not really funny. |

## B   TEXT GENERATION EXAMPLES

When performing generation thorugh the naive model with DP guarantees, we noticed that it was often unpredictable if the model would output text according to its conditional prompting. This is undesirable when generating text for a synthetic dataset, where the samples need to be generated for a particular class. We see that the output is much more consistent in our approach with the multitask model. This is evidence that separating transformer embeddings with respect to task-relevant factors enables more consistent text generation towards a desired class.

Table 7: Text Samples From SST2 Models ($\epsilon = 3$)

| Prompt | Naive Model Text ($\epsilon = 3$) | Multitask Model Text ($\epsilon = 3$) |
|---|---|---|
| BOS positive SEP | • there is something genuinely interesting about this movie – this is what it does – but if you are holding a more recent movie up.
• not as great as you could have hoped for but rather, but it leaves the essence of my greatest hour.
• i think this is one of the most gripping films of all time and makes me the most disappointed guy you have ever seen. | • the latter gives a shining, serenely sunny but pleasantly funny story with a deep sense of humor.
• an act of creative genius that builds both a wide-angle of film and a deft blend of comedy.
• it is a gripping action thriller which is a solid work of humour that hits great length and is worth checking out. |
| BOS negative SEP | • this is the most brilliant film ever, probably because it is sopositive, an absurd mix of funny tales from a writer
• the end of the film draws the line between how flawed and depressing it is.
• it is impossible not to giggle at the idea that an audience just liked the film. | • the show's biggest failure is its own unproduced,'s failure to get its audience impression, as well as his incompetence.
• the characters are often portrayed as the protagonists' parents who are misunderstandingly disappointed by the lazy and lazy characters.
• the ending has nothing to do with the world of history. |

