# OpenReview forum: "Differentially Private Conditional Text Generation For Synthetic Data Production"
_ICLR.cc/2023/Conference — Submitted to ICLR 2023_

### Official Review · Reviewer_EbEe · 2022-10-24

**Confidence:** 3
**Correctness:** 3
**Technical Novelty And Significance:** 2
**Empirical Novelty And Significance:** 2
**Recommendation:** 3

**Clarity, Quality, Novelty And Reproducibility:**

The quality and clarity are not fair enough. The novelty is fair. The experimental part seems reproducible.

**Strength And Weaknesses:**

Strength:

The paper addresses an important problem of differential privacy for language modeling. The motivation is clearly stated.

Weaknesses:

1. The paper is not well organized. The proposed method is described in the experimental part, which makes the reader confused about the motivation and the target of the paper. The main sampling strategy is based on a prior work called PPLM. However, the PPLM is not well introduced, which makes the method part hard to understand for the reviewer without prior knowledge about this method.

2. Some baselines are not mentioned and compared in the experimental part. For example, [1] also discussed the differential privacy problem for language modeling.

[1] Shi W et al. , Selective Differential Privacy for Language Modeling, NAACL 2022

**Summary Of The Paper:**

The paper proposes a synthetic data generation method for differential privacy guarantee for large language models. More specifically, the author applies the PPLM-based gradient approach with a discriminator. The proposed method is evaluate on SST-2 and AG datasets.

**Summary Of The Review:**

The paper targets to solve an important question about differential privacy for language models. However, the paper is not well organized and prepared to demonstrate the effectiveness of the proposed method.

---

### Official Review · Reviewer_hB6f · 2022-10-24

**Confidence:** 4
**Correctness:** 4
**Technical Novelty And Significance:** 2
**Empirical Novelty And Significance:** 3
**Recommendation:** 5

**Clarity, Quality, Novelty And Reproducibility:**

The paper is easy to read and understand. Proposed a multi-task learning approach to combine the language modeling and the discriminator training sumultaneously. The architecture diagram for the various approaches makes it amenable to reimplementation and potentially the code will be released in the future.

**Strength And Weaknesses:**

Pros:

   (A) They are able to generate high-fidelity and high-utility synthetic data as shown on two datasets in the experiments section.
   (B) The novel multi-task learning approach gives them the additional boost over the naive sequential approach. Also, the UMAP projections shows the clear separation between the positive and negative labels.

Cons:

   (i) There is not much novelty in terms of the proposed approach. Existing DP settings and code (RDP) are used for implementing the privacy aspects and similarly the multi-task learning is also relatively straightforward.
   (ii) There is still substantial gap between original and the synthetic versions (0.94 vs 0.89) and it would have been interesting to address that gap even before attempting the DP guarantees.


**Summary Of The Paper:**

The paper tackles the problem of sharing text data by utilizing differential privacy methods. They show how a naive approach does not give good text aligned with the class labels but their new proposed architecture does. Experiments showcasing the efficacy of their approaches are shown by using a pretrained GPT-2 model and adapted on to two datasets namely SST-2 and AG news.

**Summary Of The Review:**

Overall, a good practical paper which would be interesting for the synthetic data community. In terms of novel ideas, the impact may be modest.

---

### Official Review · Reviewer_fPMV · 2022-10-24

**Confidence:** 4
**Correctness:** 2
**Technical Novelty And Significance:** 3
**Empirical Novelty And Significance:** 1
**Recommendation:** 3

**Clarity, Quality, Novelty And Reproducibility:**

The paper's writing is somewhat unclear, with the missing caption/figure labels and the unexplained figures (as explained above under weaknesses)

The idea and proposed method are fairly novel and interesting, however, the experiments are very lacking and do not provide sufficient evidence. Also, the experiments don't seem reproducible as there are many details missing, especially under budgeting as explained above.

**Strength And Weaknesses:**

Strengths:
1.	The paper studies a very relevant problem, as synthesizing DP data (especially text) could have many applications, especially on enterprise level.
2.	The proposed method of conditioning on class label and using controllable generation is new and has not been explored before.

Weaknesses:
1. The paper lacks experiments that would help make its case and show the superiority of the proposed method, and how it is actually better, given the overheads it has. More specifically, these are the questions I have regarding experiments:
a.	Table 1, which is basically the main downstream evaluation result, seems to only show results for the proposed method, and not baselines 1 and 2. This makes this evaluation not very helpful, as we don’t know how the baselines are doing, and how much improvement we are getting from what trick. To clarify: there should be comparisons with at least one baseline, and ablations against others so we can see how much benefit a) conditioning on label b)PPLM and c) multi-task learning have. Right now we don’t even know if the proposed method does better than vanilla DP-SGD training a transformer-based classifier on the generated text. The only comparison we do get with one of the baselines is in Figure2, where the experimental setup is not even explained, but it seems to be a comparison of embeddings of the methods, using Random Forrest and is not an end-to-end downstream comparison.
b.	The overheads of the proposed method are not at all discussed. By overhead I mean any extra cost, at training or inference. For instance decoding (inference) with PPLM is actually quite expensive (due to the gradient flow and backprop, and also using a discriminator) compared to non-controllable generation. These costs are not at all discussed/explored/measured in the paper
c.	The paper has missed a very relevant related work, submix [1] which tackles the same problem of DP generation of text. I think a comparison with this paper is necessary.

2. The privacy budgeting of Baseline 2 is not at all discussed: given how this is a two stage approach, where outputs from stage 1 + input to stage 1 (the labels) go to stage 2, the privacy accounting is actually non-trivial. A conservative approach would be to compose the budgets used for stage 1 and 2, as we are re-using the labels. However I am sure better accounting could be done. This affects the reported $\epsilon$ and therefore the privacy utility.

3. Using N-gram counts as a measure of privacy is very unorthodox and uninformative as privacy is not necessarily violated if a unigram is regurgitated. Researchers usually use metrics like recall of Membership Inference Attacks [2-4], exposure metric [5], extraction attack success [6].

Minor issues:
1.	The figures explaining baseline 2 and the proposed method don’t really have captions or labels.
2.	 Figures 3 and 4 aren’t referenced anywhere in the text. I assume they relate to section 6.1?


Refs:
[1] Ginart, Antonio, et al. "Submix: Practical private prediction for large-scale language models." arXiv preprint arXiv:2201.00971 (2022).

[2] Shokri, Reza, et al. "Membership inference attacks against machine learning models." 2017 IEEE symposium on security and privacy (SP). IEEE, 2017.

[3]  Mireshghallah, Fatemehsadat, et al. "Quantifying privacy risks of masked language models using membership inference attacks." arXiv preprint arXiv:2203.03929 (2022).

[4] Carlini, Nicholas, et al. "Membership inference attacks from first principles." 2022 IEEE Symposium on Security and Privacy (SP). IEEE, 2022.


[5] Carlini, Nicholas, et al. "The secret sharer: Evaluating and testing unintended memorization in neural networks." 28th USENIX Security Symposium (USENIX Security 19). 2019.

[6] Carlini, Nicholas, et al. "Extracting training data from large language models." 30th USENIX Security Symposium (USENIX Security 21). 2021.


**Summary Of The Paper:**

This paper aims at synthesizing datasets from private data, using differential privacy. The main goal is to generate data for NLP classification tasks, and the approach this paper takes is to do so through conditional generation, as opposed to just training a generative model with DP-SGD and then taking free form samples from it. The paper discusses three main methods, two as baselines and one as the main proposed method: the first baseline is augmenting the private, training data with its class label (for instance if positive class, you pre-append with a positive token, if negative sentiment class, negative token), which means during generation you would basically prompt the model with the class label and have it generate. The second baseline improves on this by adding controllable generation (using PPLM), which uses a discriminator at decoding time to help enforce given attributes, for example sentiment. So apart from the prompt, the PPLM controllable generation would also help enforce the attribute. This setup is two stage, as in first the generator is trained and then the discriminator for PPLM is trained on top of it. Finally, they propose their method, which uses multi-task learning for training both the language generator and the discriminator at the same time, as opposed to consecutively, to help improve separability of the representations learned by the main model, thereby improving its downstream utility on the classification task.

**Summary Of The Review:**

I have provided extensive details of my summary under strengths and weaknesses, but to summarize, I find the paper's problem and approach really interesting. However, I find the presentation and the experimentation extremely lacking, and I feel like the paper needs much thorougher setup and ablations in order for the claims to have merit.

---

### Official Review · Reviewer_Hw7g · 2022-10-25

**Confidence:** 4
**Correctness:** 3
**Technical Novelty And Significance:** 2
**Empirical Novelty And Significance:** 2
**Recommendation:** 3

**Clarity, Quality, Novelty And Reproducibility:**

The organization of this paper is confused. The proposed method should not be put into the experiment section. The data processing section is not so important.

**Strength And Weaknesses:**

1. This paper simply combines existing methods (DP-SGD and GPT-2). The novel and the technical contribution to the community is limited.

2. The model performance (utility) is far from real use. If the \epsilon reaches a meaningful range (<10), the performance drops a lot compared with the baseline (in Table1).

3. The related work section misses some important papers or fails to cite some papers..
3.1. GPT-2 + DP in text generation.  SeqPATE: Differentially Private Text Generation via Knowledge Distillation. I admit this paper was only accepted by NIPS'22 just now. The authors are not required to compare this paper as a baseline. However, it's necessary to discuss the difference between your paper and this paper.

3.2. GPT-2 + DP-SGD: Xuechen Li, Florian Tramèr, Percy Liang, and Tatsunori Hashimoto. Large language models can be strong differentially private learners. In ICLR, 2022. This paper was published at ICLR'22 instead of arxiv only.

3.3.GPT-2 + DP-SGD:  Da Yu, Saurabh Naik, Arturs Backurs, Sivakanth Gopi, Huseyin A Inan, Gautam Kamath, Janardhan Kulkarni, Yin Tat Lee, Andre Manoel, Lukas Wutschitz, et al. Differentially private fine-tuning of language models. In ICLR, 2022

4. Some competitive DP-based baselines are missing . The technique proposed in [Xuechen Li' ICLR'22] and [Da Yu' ICLR'22] can be easily used in this method.

5. The organization of this paper should be improved.

**Summary Of The Paper:**

This paper propose a DP-SGD based learning algorithm to achieve synthetic data generation. The proposed method achieve DP to protect the privacy of the training data.

**Summary Of The Review:**

This paper does not meet the borderline of ICLR.

---

### Decision · Program_Chairs · 2023-01-20

**Decision:**

Reject

**Justification For Why Not Higher Score:**

There are several unsolved concerns about this paper, and there are not authors' responses.

**Justification For Why Not Lower Score:**

N/A

**Metareview: Summary, Strengths And Weaknesses:**

The paper proposes to use synthetic text-generation methods to curate private text classification datasets. In particular, the proposed approach performs conditional generation while maintaining differential privacy guarantees.

Strength:
+ The research topic is timely and interesting.


Weakness:
- Reviewres pointed out some related works are missing and the paper should position itself in a better way.
- The novelty is relatively thin.
- The experitments do not fully supprot the claim. More anlaysis and detailed disucssion are needed to make the paper more convising.



**Summary Of Ac-Reviewer Meeting:**

N/A